# Efficient Deep Reinforcement Learning Requires Regulating Statistical Overfitting

## Abstract

Deep reinforcement learning algorithms that learn policies by trial-and-error must learn from limited amounts of data collected by actively interacting with the environment. While many prior works have shown that proper regularization techniques are crucial for enabling data-efficient RL, a general understanding of the bottlenecks in data-efficient RL has remained unclear. Consequently, it has been difficult to devise a universal technique that works well across all domains. In this paper, we attempt to understand the primary bottleneck in sample-efficient deep RL by examining several potential hypotheses such as non-stationarity, excessive action distribution shift, and overfitting. We perform thorough empirical analysis on state-based DeepMind control suite (DMC) tasks in a controlled and systematic way to show that statistical overfitting on the temporal-difference (TD) error is the main culprit that severely affects the performance of deep RL algorithms, and prior methods that lead to good performance do in fact, control the amount of statistical overfitting. This observation gives us a robust principle for making deep RL efficient: we can hill-climb on a notion of validation temporal-difference error by utilizing any form of regularization techniques from supervised learning. We show that a simple online model selection method that targets the statistical overfitting issue is effective across state-based DMC and Gym tasks.

## 1 Introduction

Reinforcement learning (RL) methods, when combined with high-capacity deep neural net function approximators, have shown promise in domains such as robot manipulation (Andrychowicz et al., 2020), chip placement (Mirhoseini et al., 2020), games (Silver et al., 2016), and data-center cooling (Lazic et al., 2018). Since every unit of active online data collection comes at an expense (e.g., running real robots, chip evaluation using simulation), it is important to develop sample-efficient deep RL algorithms, that can learn efficiently even with limited amount of experience, and devising such efficient RL algorithm has been a important thread of research in recent years (Janner et al., 2019b; Chen et al., 2021; Hiraoka et al., 2021).

In principle, off-policy RL methods (e.g., SAC (Haarnoja et al., 2018b), TD3 (Fujimoto et al., 2018), Rainbow (Hessel et al., 2018)) should provide for good sample efficiency, because they make it possible to improve the policy and value functions for many gradient steps per each step of data collection. However, this benefit does not appear to be realized in practice, as taking too many training steps per each collected transition actually harms performance in many environments. Several hypotheses, such as overestimation (Thrun & Schwartz, 1993; Fujimoto et al., 2018), non-stationarities (Lyle et al., 2022), high variance (Bjorck et al., 2021a), or overfitting (Nikishin et al., 2022) have been proposed as the underlying cause with comprehensive analyses, and several mitigation strategies, such as model-based data augmentation (Janner et al., 2019a), the use of ensembles (Chen et al., 2021), network regularizations (Hiraoka et al., 2021), and periodically resetting the RL agent from scratch while keeping the replay buffer (Nikishin et al., 2022), have been proposed as methods for enabling off-policy RL algorithms to effectively use more gradient steps. While each of these methods significantly improves sample efficiency, the efficacy of these fixes can be highly task-dependent (as we will show in our experiments), and understanding what the true underlying issue that these methods address is still an important on-going research question.

In this paper, we attempt to understand why taking more gradient steps can lead to worse performance with deep RL algorithms, why heuristic strategies can help in some cases, and how this

challenge can be mitigated in a more principled and direct way. Through empirical analysis with the recently proposed tandem learning paradigm (Ostrovski et al., 2021), we show that in the early stages of training, TD-learning algorithms tend to quickly overfit on temporal-difference (TD) error (i.e., the error between the Q-network and the bootstrapping targets), reflected as a gap between the TD errors computed on the training set and a held-out validation set, and give rise to a worse final solution. We further show that many existing methods devised for the data-efficient RL setting are effective insofar as they mitigate this statistical overfitting problem. This insight gives a robust principle for making deep RL efficient: in order to improve data-efficiency, we can simply select the most suitable regularization for any given problem by hill-climbing on the validation TD error.

We realize this principle in the form of a simple active model selection method, that attempts to automatically discover the best regularization strategy for a given task during the course of online RL training. Concretely, our method, AVTD, trains several off-policy RL agents on a shared replay buffer where each agent applies a different overfitting regularizer. Then, AVTD dynamically selects the agent with the smallest validation TD error for acting in the environment. We show that this simple strategy alone often results in performance that matches or outperforms individual regularization schemes across a wide array of Gym and DMC tasks. Critically, note that unlike prior regularization methods, whose performance can vary drastically across domains, our approach provides a simple and robust method to automatically attain good performance.

To summarize, our first contribution is an empirical analysis of the bottlenecks in sample-efficient deep RL. We rigorously evaluate several potential explanations behind these challenges, and observe that statistical overfitting on the TD-error in the early stages of training is one of the biggest culprits that inhibits performance of data-efficient deep RL. Our second contribution, is a simple active model selection method (AVTD) that attempts to counteract automatically select regularization schemes by hill-climbing on validation TD-error. Our method often matches or outperforms the best individual regularization scheme across a wide range of Gym and DMC tasks.

## 2 PRELIMINARIES AND PROBLEM STATEMENT

The objective of reinforcement learning is to maximize the long-term discounted return in a Markov decision process (MDP) (Puterman, 1994), $(\mathcal{S}, \mathcal{A}, R, P, \gamma)$, consisting of a state space $\mathcal{S}$, an action space $\mathcal{A}$, a reward function $r(\mathbf{s}, \mathbf{a})$, a transition dynamics model $P(\mathbf{s}'|\mathbf{s}, \mathbf{a})$, and a discount factor $\gamma \in [0, 1)$. The Q-function $Q^\pi(\mathbf{s}, \mathbf{a})$ for a policy $\pi(\mathbf{a}|\mathbf{s})$ is the expected long-term discounted reward obtained by executing action $\mathbf{a}$ at state $\mathbf{s}$ and following $\pi(\mathbf{a}|\mathbf{s})$ thereafter, $Q^\pi(\mathbf{s}, \mathbf{a}) := \mathbb{E}_\pi \left[ \sum_{t=0}^\infty \gamma^t r(\mathbf{s}_t, \mathbf{a}_t) \right]$. The optimal Q-function is achieved when it satisfies the Bellman equation: $Q^\star(\mathbf{s}, \mathbf{a}) = \mathbb{E}_{\mathbf{s}' \sim P(\mathbf{s}'|\mathbf{s}, \mathbf{a})} \left[ r(\mathbf{s}, \mathbf{a}) + \gamma \max_{\mathbf{a}'} Q^\star(\mathbf{s}', \mathbf{a}') \right]$.

Practical off-policy methods (*e.g.*, Mnih et al., 2015; Hessel et al., 2018; Haarnoja et al., 2018a) train a Q-network, $Q_\theta$, to minimize the temporal difference (TD) error:

$$L(\theta) = \mathbb{E}_{(\mathbf{s}, \mathbf{a}, \mathbf{s}') \sim \mathcal{D}} \left[ \left( r(\mathbf{s}, \mathbf{a}) + \gamma \bar{Q}(\mathbf{s}', \mathbf{a}') - Q_\theta(\mathbf{s}, \mathbf{a}) \right)^2 \right], \tag{1}$$

where $\mathcal{D}$ is the replay buffer consisting of the transitions $(\mathbf{s}, \mathbf{a}, \mathbf{s}')$ collected so far, $\bar{Q}$ is the target Q-network that is often updated to follow the Q-network $Q_\theta$ with delay or smoothing (Fujimoto et al., 2018) so that the target does not move too quickly, and $\mathbf{a}'$ is usually drawn from a policy $\pi(\mathbf{a}|\mathbf{s})$ that can maximize or approximately maximize $Q_\theta(\mathbf{s}, \mathbf{a})$.

In theory, these off-policy algorithms can be made very sample efficient by making sure the Q-network fits the current replay buffer well, which in practice translates to taking more update steps of the Q-networks per environment step, or higher update-to-data ratio (UTD) (Chen et al., 2021). However, naïvely doing this can lead to worse performance (e.g., on DMC tasks (Nikishin et al., 2022) and on MuJoCo gym tasks (Janner et al., 2019b)).

There have been many prior methods proposed for the high UTD regime (e.g., DroQ (Hiraoka et al., 2021), REDQ (Chen et al., 2021), resets (Nikishin et al., 2022)). However, if we pick out the best methods out of these prior methods and some simple baseline regularization schemes that we study in this work (e.g., weight decay (Loshchilov & Hutter, 2017), dropout (Gal & Ghahramani, 2016), spectral normalization (Miyato et al., 2018)), none of them seems to work well across the different tasks (see Appendix A, Figure 10). What is the primary culprit that can explain the high UTD challenge? Can we address it in a more direct and principled way?

## 3 THE PRIMARY CULPRIT BEHIND FAILURE OF HIGH UTD DEEP RL

In this section, we attempt to understand the underlying causes behind the failure of off-policy RL algorithms in the high UTD deep RL and whether prior sample-efficient RL algorithms have addressed these problems appropriately. We examine several plausible hypotheses that prior works posit: Q-value overestimation due to distribution shift (Fujimoto et al., 2019a; Kumar et al., 2020), non-stationarity due to changing data distributions (Lyle et al., 2022), as well as early overfitting to the replay buffer (Nikishin et al., 2022). We first describe the setup for our empirical analysis. Then, in the subsequent section, we demonstrate through a controlled study that the aforementioned hypothesized reasons are not sufficient to explain the challenges with high UTD. Then, by demonstrating that high UTD deep RL usually results in high generalization gap in the TD error, we argue that the main culprit behind the failure mode of high UTD learning is likely statistical overfitting. We validate this hypothesis by evaluating some recently proposed regularizers, and show that these regularizers are effective insofar as they address statistical overfitting.

**Experimental setup.** We first describe the setup for our analysis. Many of the experiments in our empirical study utilize passive sources of data (i.e., operate in an "offline" regime) obtained from previous online RL runs. We replay this data in different ways to control for and examine various hypotheses. For generating this logged data, we utilize one run of a resetting SAC agent from Nikishin et al. (2022), which has been trained with a UTD value of 9. Our analysis analyzes a standard SAC agent in the high UTD regime. Since our analysis operates in the offline setting, to stabilize TD learning, we additionally normalize the features of the last layer (such normalization has also been used previously stabilize TD learning (Bjorck et al., 2021a; Kumar et al., 2021a)). We refer this as feature normalization (**FN**). We added FN in the last layer of the Q-network in all our analysis experiments except DroQ, as it already utilizes LayerNorm. For fair comparisons, we use feature normalization in all the experiments including the online setting in this section. While we keep most of the hyperparameters the same, there are still some small differences between the online and offline settings. In the online setting, we use the standard SAC which uses entropy backup in the bellman update. In the offline setting, we remove the entropy term in the bellman backup and use deterministic backup (the mean of the action from the Gaussian actor is used). Our analysis focuses on the `fish-swim` environment from DMC suite since high UTD training results in the largest gap in this domain (see the online column in Appendix B.1, Figure 11). We obtain similar trends for many other experiments; a complete set of our analysis results are in Appendix B.1. The confidence interval in our performance curves refers to the standard error computed over 8 random seeds. See implementation details about different regularizers in Appendix B.

### 3.1 CAN POOR DATA COLLECTION, DISTRIBUTION SHIFT OR NON-STATIONARITY EXPLAIN THE FAILURE OF HIGH UTD LEARNING?

First observe that, as expected, the performance of a standard SAC agent degrades as the UTD increases (Figure 1-**top**). In this section, we attempt to understand if this performance degradation can be attributed to **(a)** poor data collection, **(b)** excessive action distribution shift and overestimatioon in the value function or **(c)** non-stationarity of the replay buffer.

**(a) Quality of data used for training.** One might speculate that SAC is expected to behave poorly in the high UTD regime due to its inability to effectively collect exploratory data. To understand if this might be the primary culprit behind the poor performance from high UTD training, we analyze the behaviors of RL methods with high UTD in an *offline* setting that removes the influence of data collection strategies. If indeed the negative effects of higher UTD ratios are entirely due to exploration, we would expect this change to greatly mitigate the bottlenecks with higher UTD ratios. In this study, we trained SAC with different UTD values on the aforementioned logged dataset, but replayed the data sequentially (following the tandem learning protocol (Ostrovski et al., 2021)). This approach mimics the way a typical online RL agent would gradually observe data as it explores, but here, the dataset itself is independent of the SAC agent being trained. We refer this setting as the *offline streaming* and it still preserves the challenges pertaining to learning from data, including the effect of the training data distribution and data quantity. As shown in Figure 1-**middle**, the performance of SAC still degrades as UTD increases, even though it is being trained on data from a well-trained agent. This suggests that the poor data collection alone does not explain the failure of high UTD learning.

**(b) Non-stationarity in replay buffer data distributions.** Another potential explanation for the challenges in learning with higher UTD values is non-stationarity: with higher UTD, the algorithm makes more gradient updates on the learned policy, such that the distribution of data collected by the policy would change more drastically between iterations of learning. Sudden changes in the data distribution and non-stationary target values have been regarded as challenges in online RL (Igl et al., 2020a). As before, we construct an experiment that removes non-stationarity of data. To do so, we rerun SAC with different UTDs in the offline streaming setting discussed above, but now also reshuffle the buffer before training SAC. That is, while the streaming setting replays the data in the order it was collected by the online RL agent that was used for collecting the logged dataset, this new setting presents data sequentially, but not in the order that it was collected. This ensures that the data *distribution* of samples used for training is stationary and does not change over the course of training. We refer this as the *offline shuffled streaming* setting. Note however that the underlying RL algorithm still observes new data points as it trains for longer. Figure 1-**bottom** shows that a similar performance trend still holds in this shuffled streaming setting, indicating that non-stationarity of the data distribution alone also does not explain the failure of high UTD learning.

**(c) Distribution shift and out-of-distribution (OOD) actions.** Our analysis so far suggests that the challenges in learning with high UTD are related to being able to effectively learn from data: even when the data quality and non-stationarity are accounted for, the performance with high UTDs is worse. One might speculate that an obvious challenge for learning from data is action distribution shift or OOD actions (Fujimoto et al., 2019b; Levine et al., 2020; Kumar et al., 2019): higher UTDs require more off-policy Bellman backups, resulting in backups from OOD actions, and Q-value overestimation (Thrun & Schwartz, 1993; Van Hasselt et al., 2016; Fujimoto et al., 2018). To investigate this, we plot the gap in Q-values at actions chosen by the policy and the actions in the dataset: $\Delta Q = \mathbb{E}_{\mathbf{s} \sim \mathcal{D}, \mathbf{a}^\pi \sim \pi(\mathbf{a}|\mathbf{s})}[Q_\theta(\mathbf{s}, \mathbf{a}^\pi)] - \mathbb{E}_{\mathbf{s}, \mathbf{a}^\beta \sim \mathcal{D}}[Q_\theta(\mathbf{s}, \mathbf{a}^\beta)]$ which provides an upper-bound on the overestimation, and find that this value is roughly identical for all UTD values we consider (Figure 2).

This suggests that higher UTDs do not lead to any more overestimation than the smaller UTD values that work well, and hence the performance degradation from ramping up the UTD cannot be explained by overestimation due to action distribution shift. Note that we are not claiming that action distribution shift is not a problem in general, or that in will not happen at all, but that our evidence shows that in the online RL settings with high UTD that we study, overestimation due to distributional shifts is not the primary culprit.

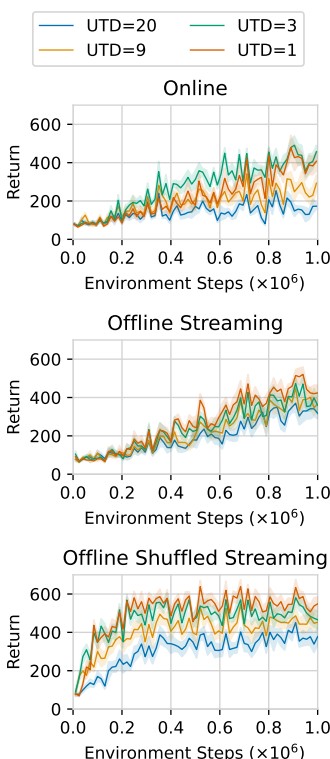

Figure 1: The effects of varying UTD ratios on the performance of SAC agents augmented with feature normalization on `fish-swim` task under online (**top**), offline streaming (**middle**), and offline shuffled streaming (**bottom**) settings.

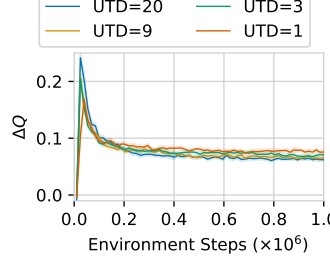

Figure 2: (**Shuffled Streaming**): The $\Delta Q$ on `fish-swim` with varying UTD ratios. Typical offline RL issue of being over-optimistic on OOD actions does not appear in the regime we study.

### 3.2 CAN STATISTICAL OVERFITTING EXPLAIN THE FAILURE OF HIGH UTD LEARNING?

Even after using data of high quality obtained from the run of periodically resetting SAC, correcting for non-stationarity and distribution shift, we find that the challenges with high UTD RL still remain. This hints at the possibility that the actual underlying issue is statistical in nature, such as some form of overfitting to the data points that were used for training. This sort of statistical overfitting should be identifiable by measuring errors on a held-out validation dataset. In particular, we plot the training and validation TD errors as well as the standardized TD gap (*STD gap*), defined

as: $\Delta_{\text{STD}} = \frac{L_{\text{train}} - L_{\text{valid}}}{L_{\text{valid}}}$. We observe in Figure 3, that the STD gap tends to be correlated with failure cases with high UTD. Note that both validation TD error and the STD gap correlate well with the increase of UTD ratio, demonstrating that the degrade in performance is likely due to statistical overfitting in this environment. We also remark that our hypothesis of statistical overfitting is distinct from prior works: while Nikishin et al. (2022) utilizes the term "overfitting" to collectively refer to the challenges in high UTD learning, this prior work does not use a held-out validation set for their analysis, which makes it impossible to conclude if the challenges are due to statistical overfitting on the samples in the training dataset.

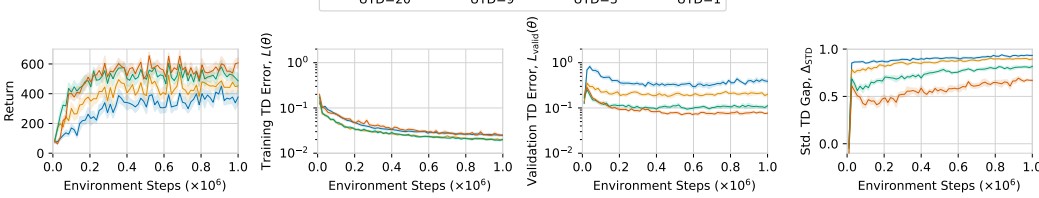

Figure 3: **(Shuffled Streaming) Statistical overfitting appears on `fish-swim`.** Higher UTD leads to heavier overfitting which is reflected on lower training TD error (**second plot**), higher validation error (**third plot**), and lower performance degrades (**first plot**). The generalization gap (**right**) quantifies the amount of overfitting each UTD setting experiences. Higher UTDs (UTD=9/20) tend to have higher generalization gaps. All agents use feature normalization in the last layer to stabilize TD learning. The evaluation of the TD error is done on the growing training/heldout replay buffer (originally collected by the online SAC agent that resets periodically during training).

**Analysis on other DMC environments.** We have shown empirical evidence that statistical overfitting is the most plausible explanation for the failure case of high UTD learning on `fish-swim`, but how about other environments? It turns out that most other DMC tasks that suffer from the high UTD learning issue also has the same trend (see Figure 14 in Appendix B.1). This suggests that the biggest challenge that needs to be handled in such data-efficient deep RL settings is that of statistical overfitting on the TD error.

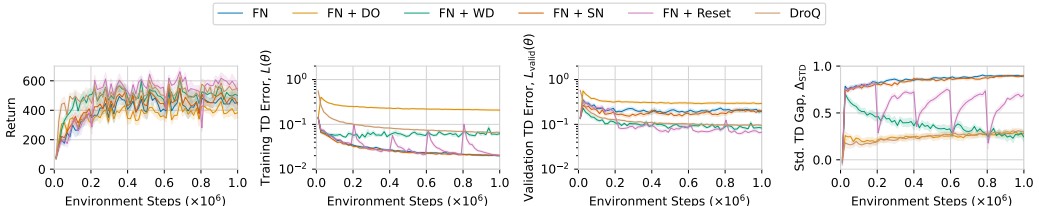

Figure 4: **Offline Shuffled Streaming** Diagnostic analysis on `fish-swim`. Regularization approaches achieve similar effects (to different degrees), alleviating overfitting and leading to lower generalization gap (**right**). All agents use feature normalization in the last layer to stabilize TD learning.

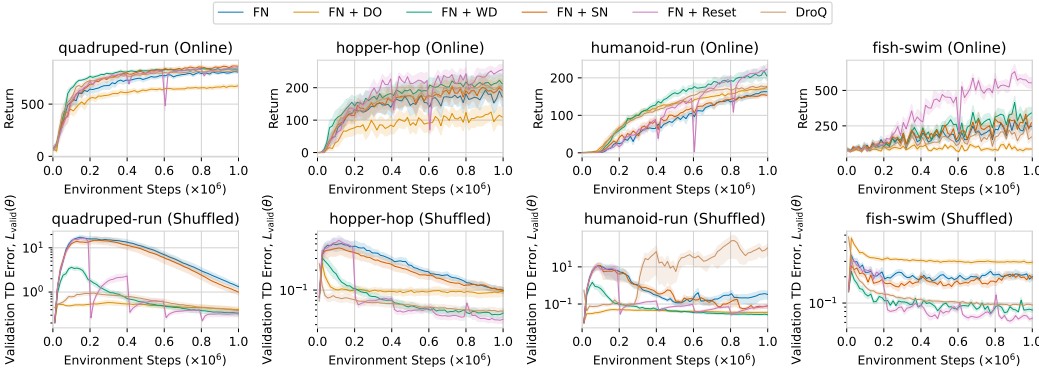

Figure 5: **The effect of regularization approaches on the online performance on TD validation error.** On `quadruped-run`, `hopper-hop`, `humanoid-run`, and `fish-swim` the performance improvements correlate well with the validation TD error in the offline setting. We include a more complete set of results and more discussions for some failure cases in Appendix B.1, Figure 12).

### 3.3 IS MITIGATING STATISTICAL OVERFITTING ABLE TO EXPLAIN THE GOOD PERFORMANCE OF PRIOR REGULARIZERS?

We provided empirical evidence in the previous section that statistical overfitting is the most plausible explanation for the failure of naïve RL methods with high UTD ratios, compared to a number of other previously hypothesized explanations. In this section, we attempt to understand if the performance improvements from a variety of previously-proposed regularizers in high UTD RL can be attributed to their effectivness in mitigating statistical overfitting. We note that none of the methods we study enable high UTD learning across all tasks (as we have previously discussed), so our study instead focuses on understanding whether methods that work *well* in each setting *also* overfit less statistically. These regularizers include dropout (Gal & Ghahramani, 2016) (**DO**, used by Hiraoka et al. (2021)), weight decay (Loshchilov & Hutter, 2017) (**WD**, used by Lillicrap et al. (2015)), spectral normalization (Miyato et al., 2018) (**SN**, used by Gogianu et al. (2021); Bjorck et al. (2021b)), periodic resets (Nikishin et al., 2022), and a combination of LayerNorm (Ba et al., 2016) and dropout (**DroQ** (Hiraoka et al., 2021)). All of these regularizers operate differently: dropout injects stochasticity into the Q-network, weight decay controls the parameter norm; Spectral normalization controls the maximum singular value of the weight matrix. We observe that SAC exhibits lower validation TD errors when trained with these regularizers in the high UTD regime (as shown in Figure 4 for `fish-swim` and Appendix B.1, Figure 16 for other environments). This further highlights that a reduction in statistical overfitting on the TD-error does correspond to better performance. We would also highlight that the regularizers that achieve the lowest validation TD error offline are usually one of the top performing methods online (Figure 5), hinting that the issues that these regularizers addressed could be mostly statistical overfitting.

## 4 DYNAMIC MODEL SELECTION BASED ON VALIDATION TD ERROR (AVTD)

The performance of various regularization methods indicates that no single regularization approach performs well on all the tasks possible, and more so, it is unreasonable to expect that a single regularizer would perform well on every deep RL problem. However, if we can devise a general principle that allows us to select from among regularization approaches, we would expect such an approach to perform well given a broad set of regularization methods it choose from. In the previous section, we observed that the validation TD error of different regularization approaches correlates well with performance in the offline setting. Can we somehow use this correlation to our advantage and select the best regularization approach automatically?

---

**Algorithm 1** AVTD

---

1: **Input: A collection of off-policy RL agents** $\{(Q_\theta^1, \pi_\theta^1), \cdots, (Q_\theta^K, \pi_\theta^K)\}$, **greedy exploration coefficient** $\varepsilon$.
2: $\mathcal{D} \leftarrow \emptyset$
3: **for** each environment step **do**
4:      With probability $\varepsilon$, $j \leftarrow \arg\max_i L(\theta_i; \mathcal{D}_{\text{heldout}})$. Otherwise, $j \leftarrow \text{Unif}(\{K\})$
5:      Sample action $a$ from $\pi_\theta^j$ and use it to act in the environment
6:      Add the new transition in the replay buffer: $\mathcal{D} \leftarrow \mathcal{D} \cup \{s, a, s', r\}$
7:      **for** $i = 1 \cdots k$ **do**
8:          Update $Q_\theta^i$ and $\pi_\theta^i$ using the replay buffer $\mathcal{D}$
9:      **end for**
10:     After every $n_{\text{episode}}$ episodes, collect a heldout trajectory and add to $\mathcal{D}_{\text{heldout}}$ with the same action selection strategy above for $\mathcal{D}$.
11: **end for**

---

A naïve approach that directly follows from our analysis would train multiple independent agents with different regularizers in parallel, for a small number of initial steps, then select the one with the smallest validation TD error and use it for the rest of training. While intuitive, this approach may not necessarily work: TD error depends on the scale of the reward function, and typically as an online RL agent makes progress towards maximizing reward and observes higher reward value, TD-error increases. This means that this naïve approach will select the agent that has made the least progress towards maximizing reward as it is likely to be the one that attains the smallest TD-error, which is not desirable. To address this shortcoming, we consider a simple modification of this idea: we

instead train multiple agents with different regularizers on a *shared* replay buffer, such that the data collection does not confound the evaluation of TD error. At each environment step, we pick the agent with the lowest validation TD error to take actions in the environment. Essentially, all the agents are learning on the same buffer, similar to the offline streaming setting, except that the active agent (the agent that is taking action currently) collects the data that goes into the replay buffer. As we will show in our experiment, this does not reduce the correlation between the validation TD error and the performance, and this selection strategy can reliably select the best performing algorithm without incurring the additional sample complexity that might result from, for example, running multiple learners in sequence to pick the best one. An overview of the algorithm is shown in Algorithm 1.

## 5 EXPERIMENTAL EVALUATION OF AVTD

The goal of our experiments is to validate the principle that hill-climbing on validation TD error to mitigate statistical overfitting can improve performance in data-effcient deep RL. To this end, we evaluate our active model selection method, AVTD along with previously-proposed regularization strategies for comparisons. Through experiments, we will establish that automatically selecting the regularization strategy (or strength) via AVTD is able to match or outperform the best individual stratgy. Concretely, we will answer the following questions: **(1)** Is AVTD able to select the best regularization coefficient online?, **(2)** Does AVTD still work if it were to select based on the training TD error?, and **(3)** Does AVTD match or improve the performance of the best performing regularization approach across a wide range of tasks? We first present answers to questions **(1)** and **(2)** and then present our final results in **(3)**. Implementation details are in Appendix B.

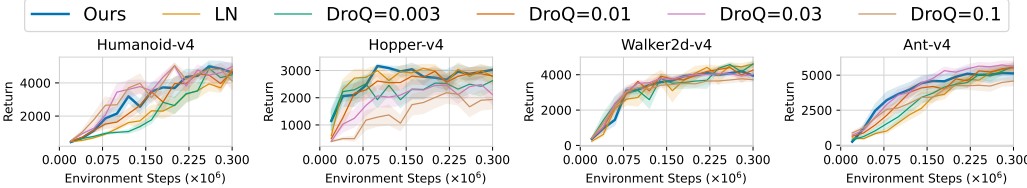

Figure 6: **Is AVTD able to select the best regularization strength in the ensemble?** We plot the evaluation return of AVTD against the evaluation return of each of the five agents trained independently. The five agents we are considering are DroQ (Hiraoka et al., 2021) with different dropout rate: 0.1, 0.03, 0.01, 0.003 and 0.0 (denoted as LN). AVTD consistently matches the top performing regularizer, suggesting that AVTD can indeed select the best regularization strength. AVTD use a greedy exploration coefficient of $\varepsilon = 0.1$.

**(1) Is AVTD able to select the best regularization strength online?** To answer this question, we use five DroQ agents with different dropout rates $(0.003, 0.01, 0.03, 0.1, 0.0)$ and see if AVTD is able to select the best dropout rate for each Gym task. We show in Figure 6 that AVTD can reliably match the performance of the best regularizer on the four Gym tasks we train on. One might argue that simply training an ensemble of these agents could achieve a similar effect. We show that this is not the case in Figure 7 by comparing to the uniform selection strategy where a randomly selected agent is used to act in the environment for any given rollout.

**(2) Is there any benefit to specifically using *validation* TD error in AVTD?** To answer this question, we performed a study where we automatically adjust the regularization strategy by hill-climbing on the training TD-error instead of the validation TD error. On fish-swim, we observe that utilizing validation TD error is critical, and training TD error leads to worse performance (Figure 8). Qualitatively, we observe that the regularization strategy selected by the hill-climbing on training TD error is the one that does not add any regularization, resulting in worse performance. This demonstrates that the principle of hill-climbing on the validation TD error is more robust than hill-climbing on the training TD error, further corroborating the insights from our empirical analysis.

**(3) Can AVTD match or exceed the performance of individual regularization strategies, aggregated across a wide range of tasks?** To evaluate overall performance of AVTD in comparison with each individual regularizer, we evaluate AVTD and all prior methods on 9 DMC tasks and 4 MuJoCo Gym tasks. The comparative evaluation of AVTD, prior works, and our weight decay baseline is shown in Figure 9. For AVTD, we use a combination of five regularization strategies: Layer-Norm, LayerNorm + WD with a weight of $0.01$, WD with a weight of $0.01$ alone, and LayerNorm + Dropout with fractions of $0.03$ or $0.01$. Including agents with the DroQ setup allows AVTD to work well on Gym tasks. Including agents with weight decay allows AVTD to work well on DMC

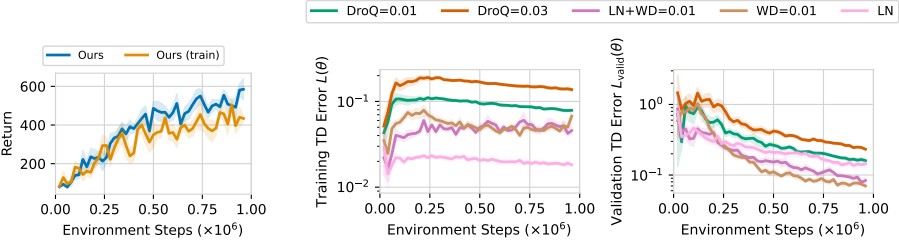

Figure 7: **Is the benefit of AVTD purely explained by training an ensemble of agents and not the use of validation error for model selection?** In this experiment, we experiment with $\varepsilon = 1.0$, which corresponds to randomly selecting agents with 100% probability. This is denoted as **Random** and the evaluation return of each agent in the ensemble is shown separately. The five agents we are considering are DroQ (Hiraoka et al., 2021) with different dropout rate: 0.1, 0.03, 0.01, 0.003 and 0.0. **Ours** shows the evaluation return of the agent that achieves the lowest validation TD error. We see that AVTD can consistently outperform uniform selection, suggesting that the benefit of AVTD is not purely coming from training an ensemble. AVTD uses $\varepsilon = 0.1$.

Figure 8: **Comparison of AVTD with the model selection based on training TD error on `fish-swim`.** **Left**: evaluation return. **Right**: validation TD error of each agent in AVTD (right) and training TD error of each agent in AVTD (train) (mid). AVTD consistently picks the agent that achieves the best performance (**WD=0.01**) whereas AVTD (train) consistently picks the agent that overfits the most. As a result, AVTD (train) achieves a lower evaluation return compared to AVTD. AVTD uses $\varepsilon = 0.1$.

tasks. The results show that for each of the tasks, there exists some prior regularization strategy that works well. However, no single strategy performs well across all the tasks. AVTD frequently matches the best performing method (with the exception of the DMC `acrobot-swingup` task) and, in the case of the hardest task (`humanoid-run`), exceeds the performance of prior methods, indicating that selecting the regularizer based on validation error is effective single approach that consistently attains good performance. In our experiments, we also include a simple, but strong baseline (**WD=0.01**) and find it to work well in general, but worse than AVTD.

## 6 RELATED WORK

**Overfitting in deep RL.** In image-based RL domains, many prior works (Song et al., 2019) have identified the overfitting issues where proper data augmentation techniques can improve performance greatly (Kostrikov et al., 2020; Laskin et al., 2020; Yarats et al., 2021; Raileanu et al., 2021). Cetin et al. (2022) identified a specific self-overfitting issue that is caused by TD learning with a convolutional encoder and low magnitude rewards. This is different from our work as we mainly focus on state-based environments with mostly dense rewards whereas Cetin et al. (2022) studies the image-input environments that have low-magnitude sparse rewards. Overfitting is also heavily studied in the offline RL setting (Kumar et al., 2021b; Arnob et al., 2021; Lee et al., 2022), and while we do run some analysis experiments in the offline setting, we follow the tandem learning protocol (Ostrovski et al., 2021), where the offline dataset is generated via an active RL agent and does not come from an arbitrary distribution. In the sample-efficient deep RL setting that we study in this paper, Nikishin et al. (2022) observed that forcing the TD learning to fit on a small initial replay buffer with a large number of gradient steps can significantly hinder the learning progress later on in the training. This prior work speculates that this observation could be due to some "overfitting-like" phenomenon, however, it does not characterize what this overfitting precisely means in this context and shows no evidence of statistical overfitting. Note that that this prior work does not utilize any notion of a held-out validation set, which we believe is essential for quantifying any form of statistical overfitting. Finally, our analysis also examines the feasibility of various other hypotheses (e.g., non-stationarities of the replay buffer (Lyle et al., 2022; Igl et al., 2020b), action distribution shift (Fujimoto et al., 2019a; Kumar et al., 2020), value under/over-estimation (Fujimoto et al., 2018;

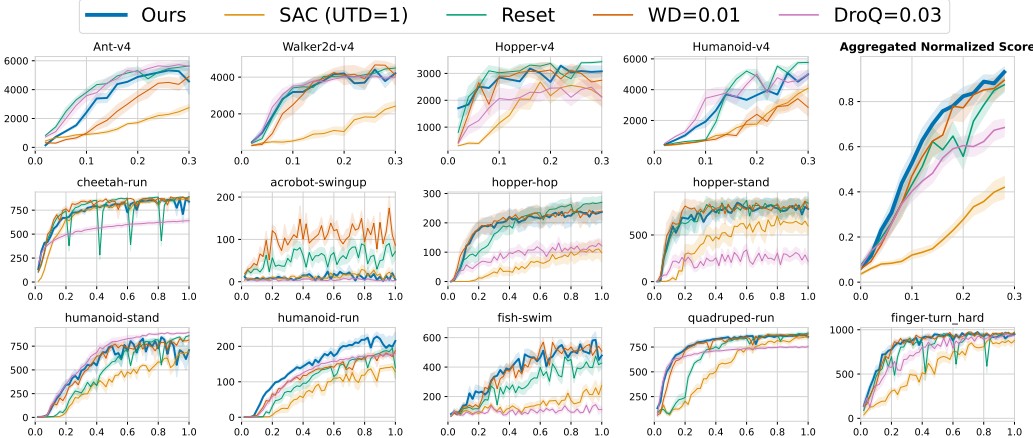

Figure 9: **AVTD comparing to the Performance of various sample-efficient RL methods based on SAC on Gym (top row) and DMC tasks (bottom two rows). SAC**: the standard SAC agent trained with UTD ratio of 1; **Reset**: the standard SAC agent trained with high UTD ratios (9 on DMC and 20 on Gym) and resets periodically (after every 200K/100K steps on DMC/Gym tasks); **WD=0.01**: the standard SAC agent trained with the high ratios and regularized with weight decay in the Q-network; **DroQ** (Hiraoka et al., 2021). AVTD performs more reliably across the board, often matching the top performing method on each environment. We use representative baselines in this plot; See Appendix B.2, Figure 18 for the full results with all baselines and the aggregated normalized score computation protocol.

Chen et al., 2021; Wang et al., 2021)) towards explaining the challenges of data-efficient deep RL, but we find that statistical overfitting is the primary culprit.

**Regularization in Deep RL.**Regularization schemes, such as Dropout (Gal & Ghahramani, 2016), LayerNorm (Ba et al., 2016), or BatchNorm (Cheng et al., 2016) have been effective in improving the sample efficiency of deep RL algorithms. For example, Hiraoka et al. (2021) uses Dropout and LayerNorm on top of SAC to attain near state-of-the-art performance on MuJoCo gym Liu et al. (2019) found that $L_2$ weight regularization on actors can improve both on-policy and off-policy RL algorithms. The solution of (Nikishin et al., 2022) which proposes periodic resets of critic weights can also be interpreted as a form of regularization. Despite the empirical successes of these regularization approaches, the understanding of the principle behind these regularization approaches is lacking. Our analysis sheds light on the connection of these regularization approaches to statistical overfitting. We also observe that the efficacy of these regularizers is quite domain dependent: not all regularizers work in all domains (see Figure 10). On the other hand, our active model selection approach, which attempts to automatically adjust the regularization scheme by hill-climbing on validation error provides a simple scheme to attain match or outperform the individual best regularizer, for *all* domains. To clarify, our approach does not propose yet another regularizer, but a model selection to select from among regularizers. Our method is also related to theoretical algorithms that attempt to do online model selection (Foster et al., 2019; Lee et al., 2021; Cutkosky et al., 2021), that study this problem from a theoretical perspective. (Khadka et al., 2019) also utilizes multiple learners and actively selects from them based on a statistical estimate of return, which is distinct from validation TD-error AVTD utilizes.

## 7 DISCUSSION

In this work, we attempted to understand the primary bottlenecks in data-efficient deep RL. Through a rigorous empirical analysis, we showed that poor performance in high UTD deep RL is often correlated with statistical overfitting, and the effectiveness of many existing regularizers can be explained by their ability to address overfitting. We use this experimental design to devise a principle for obtaining sample-efficient deep RL: by targeting this overfitting issue by an active model selection strategy, that automatically adjusts regularization based on validation TD-error, we can often match or outperform existing regularizers on each task, achieving better overall performance. While AVTD can work well across a number of domains, several important questions remain. For instance, it is not clear why and when certain regularization strategies work better than others. If we can answer this question, we can optimize for validation TD error in a more straightforward fashion without requiring multiple parallel agents. This likely would require understanding the learning dynamics of TD-learning, which is an interesting topic for future work.

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

# Appendices

## A FAILURE CASE OF EXISTING REGULARIZERS

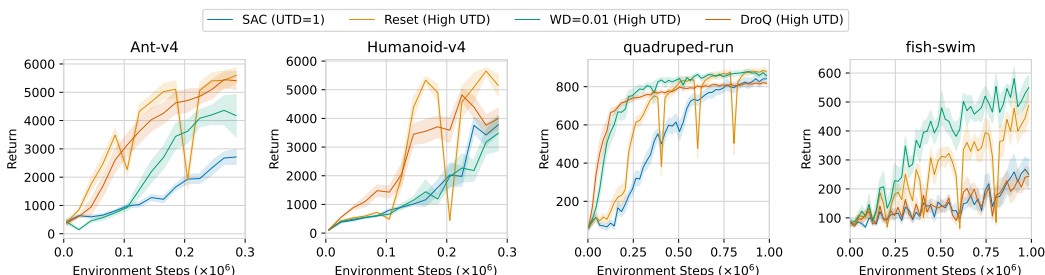

Figure 10: **Failure cases for common sample-efficient RL methods across DMC and MuJoCo gym benchmark. DroQ** is the state-of-the-art method on gym tasks, **Reset** is one of the top performing algorithm on DMC from Nikishin et al. (2022) that utilizes resets, and **WD=0.01** is a simple regularization baseline that we studied in our work that utilizes weight decay on the Q-network. While DroQ and Resets attain good performance on the Gym tasks, they perform poorly on the other set of tasks from the DMC suite. In contrast, weight decay performs well on the DMC tasks attains poor performance on Gym.

## B IMPLEMENTATION DETAILS

**AVTD**   In all our experiments, we use $\varepsilon = 0.1$ unless specified otherwise. This means that at each environment step, there is a 10% chance that a random agent is picked and the action is sampled from that random agent.

**Weight decay (WD).**   For all our experiments with weight decay, we apply AdamW (Loshchilov & Hutter, 2017) on the weight matrices of the Q-network (not on bias) except the last layer of the network (that maps the last layer feature to a scalar). When LayerNorm and weight decay are used together, weight decay is not applied on the bias and the scale learned in the LayerNorm.

**Spectral normalization (SN).**   For spectral normalization, we follow the implementation of

**LayerNorm (LN).**   For all our experiments with LayerNorm (Ba et al., 2016), we use it right before each ReLU activation in the Q-network and learns additional per-feature-element scales and biases.

**Feature Normalization (FN).**   For all our experiments with feature normalization, we use it right before the last layer of the Q-network where it is parameterized as

$$Q_\theta(\mathbf{s}, \mathbf{a}) = \frac{w^\top f_\theta(\mathbf{s}, \mathbf{a})}{\|f_\theta(\mathbf{s}, \mathbf{a})\|_2}$$

where $w$ is the last layer weight of the Q-network and $f_\theta(\mathbf{s}, \mathbf{a})$ gives the post-activation feature right before the last layer. This trick has been applied in many prior works to improve the stability of TD learning (e.g., (Bjorck et al., 2021a; Kumar et al., 2021a)).

**Dropout (DO).**   For all our experiments with dropout, we apply it in the Q-network before the ReLU activation (before LayerNorm when combined together, e.g., DroQ (Hiraoka et al., 2021)).

**Reset.**   For our experiments with Reset (Nikishin et al., 2022), we use the same strategy as the original paper where we re-initialize the agent from scratch periodically while keeping the replay buffer. For DMC tasks, we use a reset frequency of 200K steps (same as the original paper). For gym tasks, we use a reset frequency of 100K steps. To our surprise, resetting every 100K could already match the performance of DroQ (Hiraoka et al., 2021).

**Gym experimental setup.** For all the experiments on Gym tasks, we follow DroQ (Hiraoka et al., 2021) where we update the actor once per every 20 critic update steps and run 300K environment steps. We use a warmup period of 5000 steps where random actions are taken before updating the agents.

**DMC experimental setup.** For all the experiments on DMC tasks, we use a UTD ratio of 9, warmup period of 10000 steps where random actions are taken before updating the agents, and run 1M environment steps.

**SAC.** For the SAC implementation used in this paper, we build our code on top of the `jaxrl` codebase: `https://github.com/ikostrikov/jaxrl`. The hyperparameter used for SAC is attached as follows (see Table 1):

| | | |
|---|---|---|
| **Initial Temperature** | | 1.0 |
| **Target Update Rate** | update rate of target networks | 0.005 |
| **Learning Rate** | learning rate for the Adam optimizer | 0.0003 |
| **Discount Factor** | | 0.99 |
| **Batch Size** | | 256 |
| **Network Size** | | $(256, 256)$ |
| **Warmup Period** | # of initial random exploration steps | 10000 for DMC, 5000 for gym MuJoCo |

Table 1: Hyperparameters used for the SAC algorithm (Haarnoja et al., 2018b)

## B.1 DMC ANALYSIS

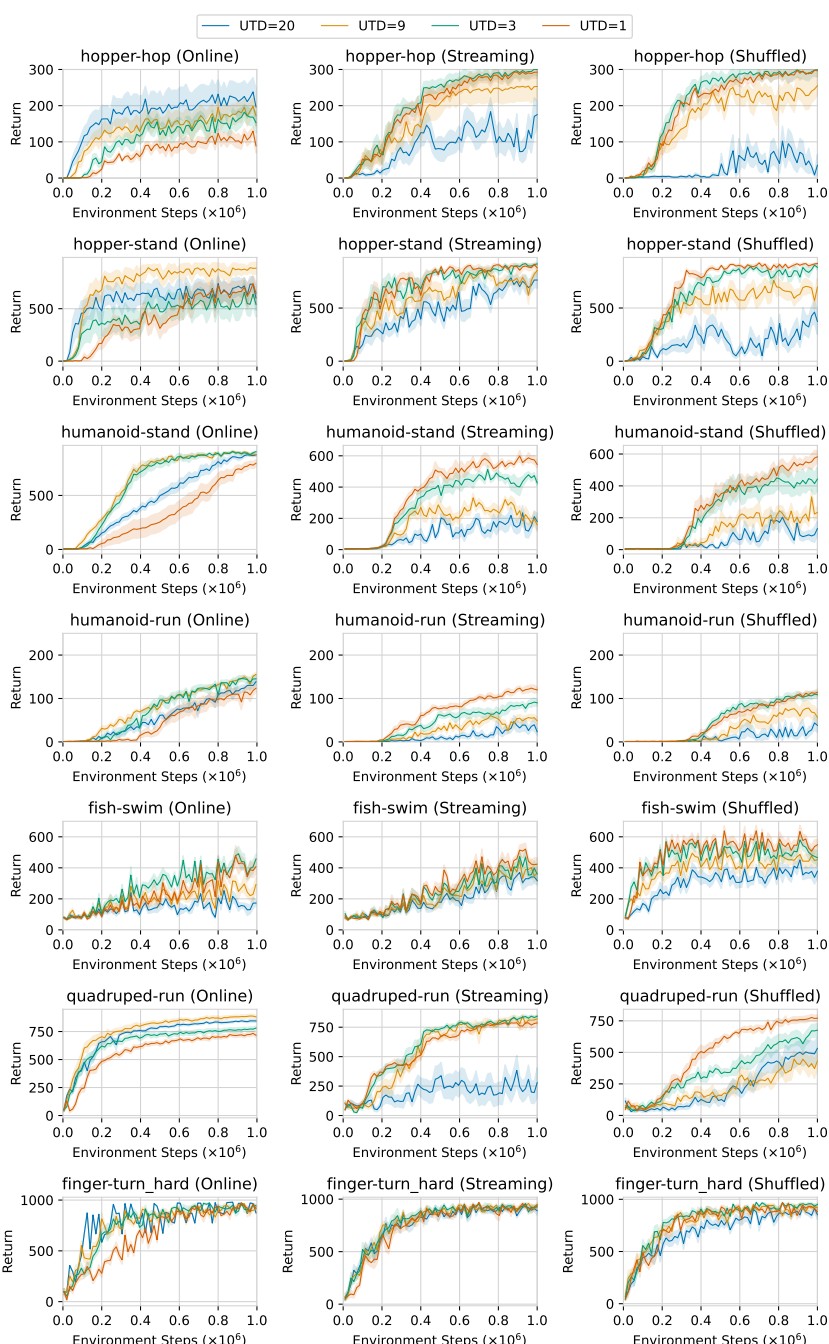

Figure 11: **The effects of UTD ratios on 7 DMC tasks**. All agents use feature normalization in the last layer feature to stabilize TD learning. Among all the tasks considered, almost all tasks exhibit the failure mode of high UTD in the online setting except on `hopper-hop` where UTD=20 performs the best. In the offline setting, the performance degrade trend is cleaner where the agents trained with UTD=1 performs the best across the board (except on `hopper-hop` on the streaming setting, `humanoid-run` on the shuffled streaming setting, and `finger-turn_hard` on both offline settings.) and the agents trained with UTD=20 performs the worst across the board (except on `quadruped-run` on shuffled streaming setting and `finger-turn_hard` on both offline settings).

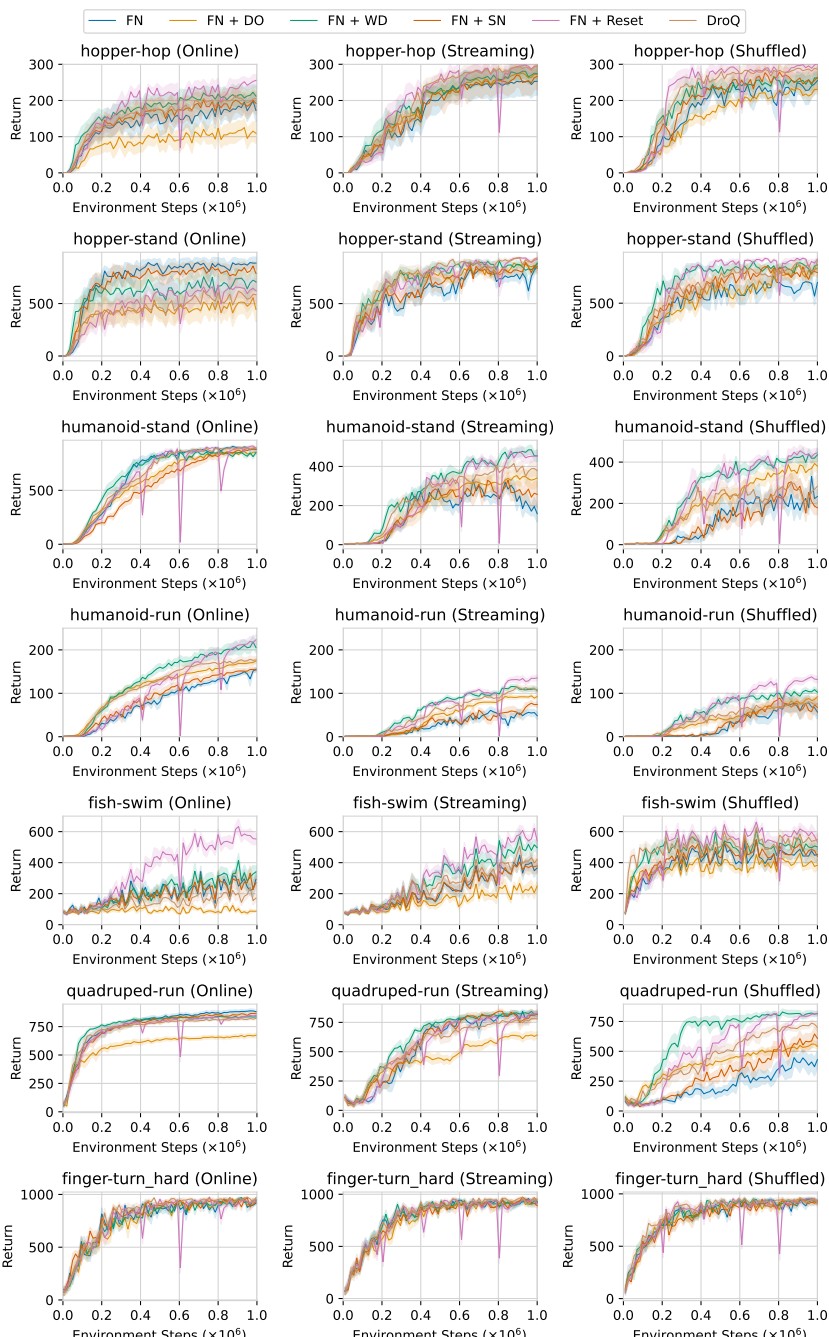

Figure 12: **The effects of various regularization approaches on 7 DMC tasks**. Among all the tasks considered, `fish-swim`, `humanoid-run` and `hopper-hop` have the most similar relative performance ordering. On `humanoid-stand`, all the other methods except the base SAC + FN correlates have similar performance ordering across online/offline settings. On `quadruped-run`, we observe that the performance gap is much bigger in the offline settings, but the relative ordering of DO, WD, Reset and DroQ is roughly preserved (FN and FN + SN are much worse in the shuffled streaming setting). The ordering is the most ambiguous in `hopper-stand`.

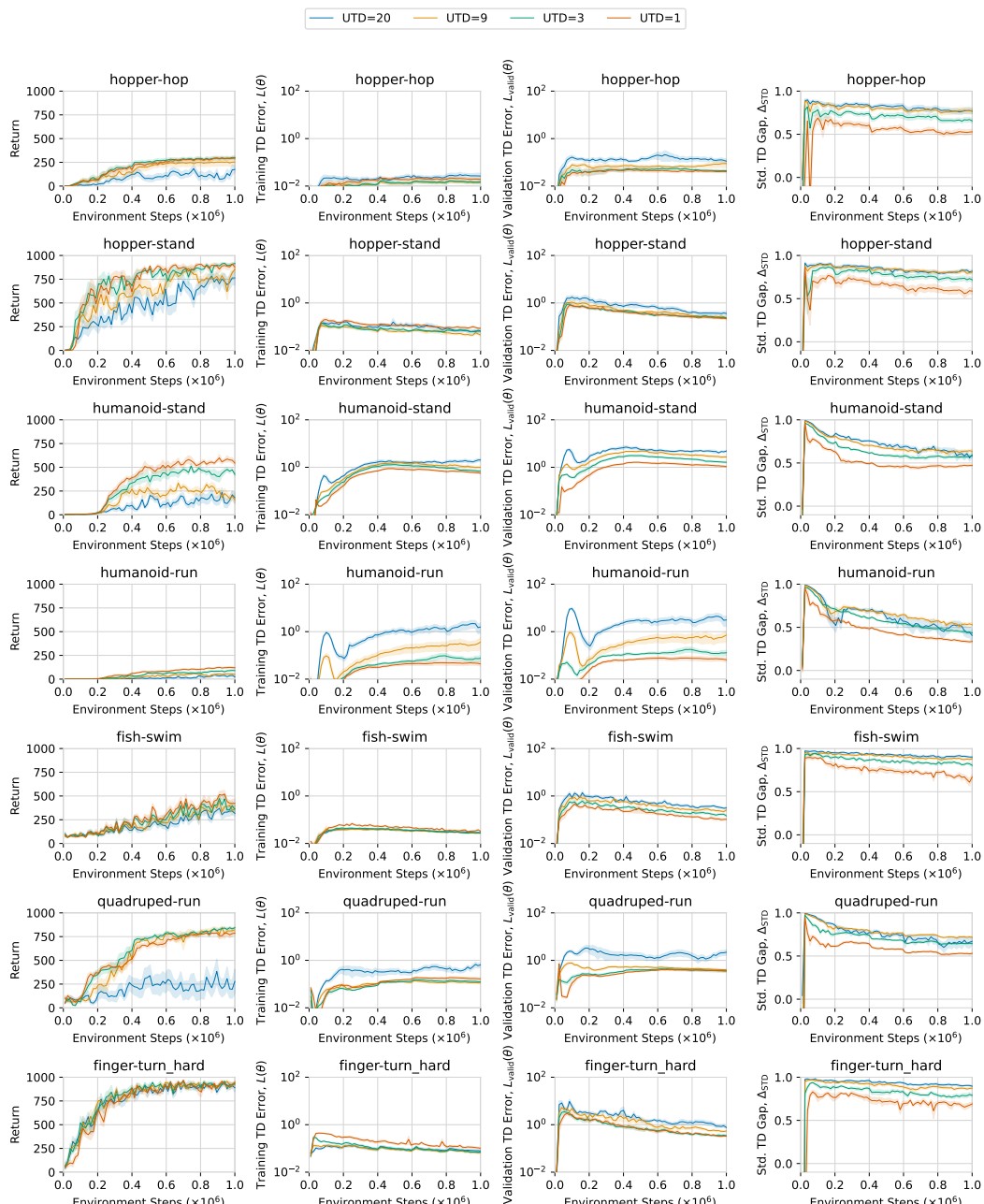

Figure 13: **The effects of UTD ratio on 7 DMC tasks in the offline streaming setting.** All agents use feature normalization in the last layer to stabilize TD learning. For all tasks except quadruped-run (where the agent trained with UTD=1 is doing a bit worse than other agents with UTD=3 and UTD=9 but achieving lower validation TD error) and finger-turn_hard (where performance does not seem to matter among different UTD ratios), the TD validation error correlated well with performance.

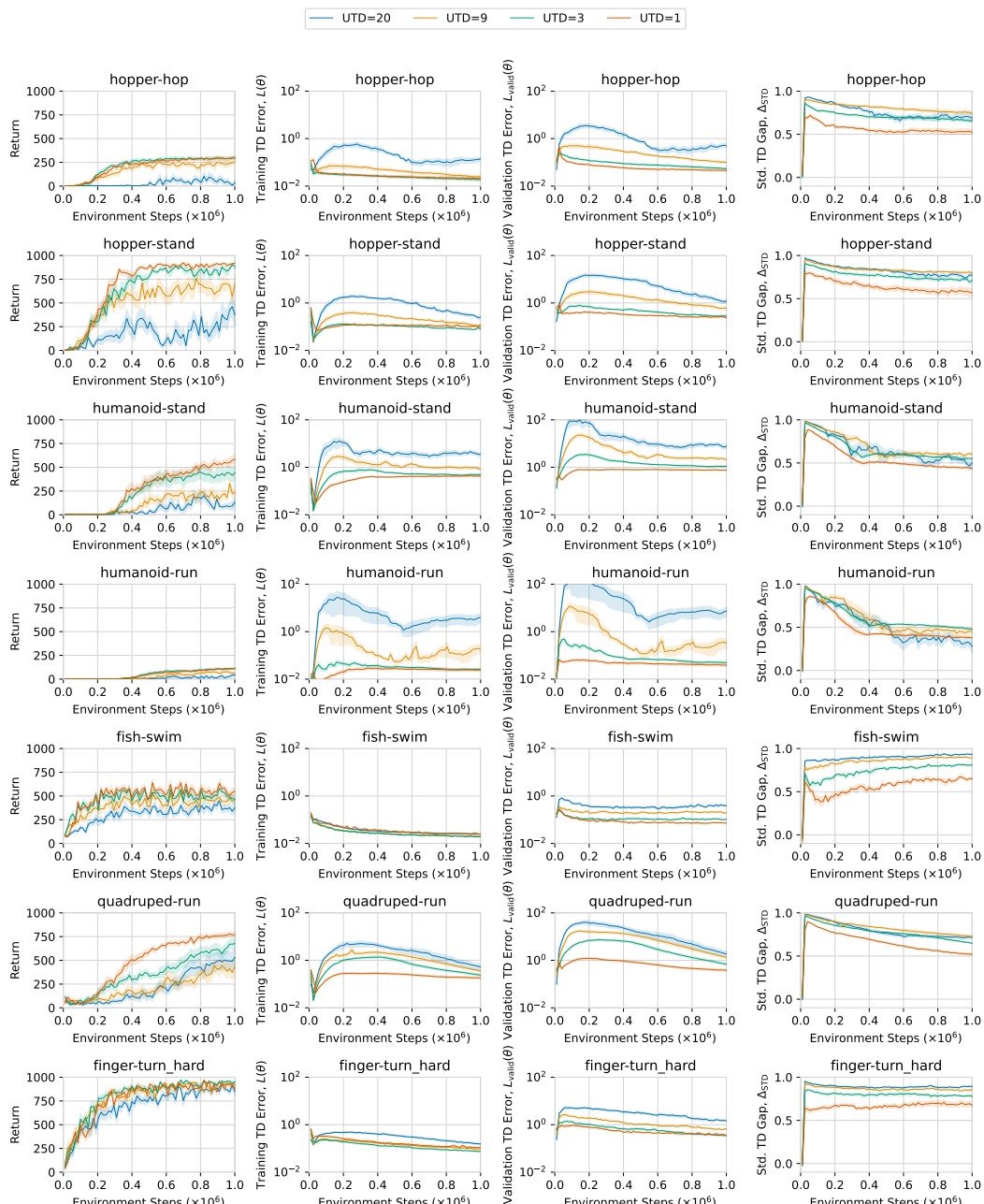

Figure 14: **The effects of UTD ratio on 7 DMC tasks in the offline shuffled streaming setting.** All agents use feature normalization in the last layer to stabilize TD learning. For all tasks except finger-turn_hard (where the performance of the agents trained with UTD=1,3,9 are indistinguishable while the validation TD errors are) and hopper-hop (where the performance of the agents train with UTD=1,3 are indistinguishable while the validation TD errors are), the TD validation error correlated well with performance.

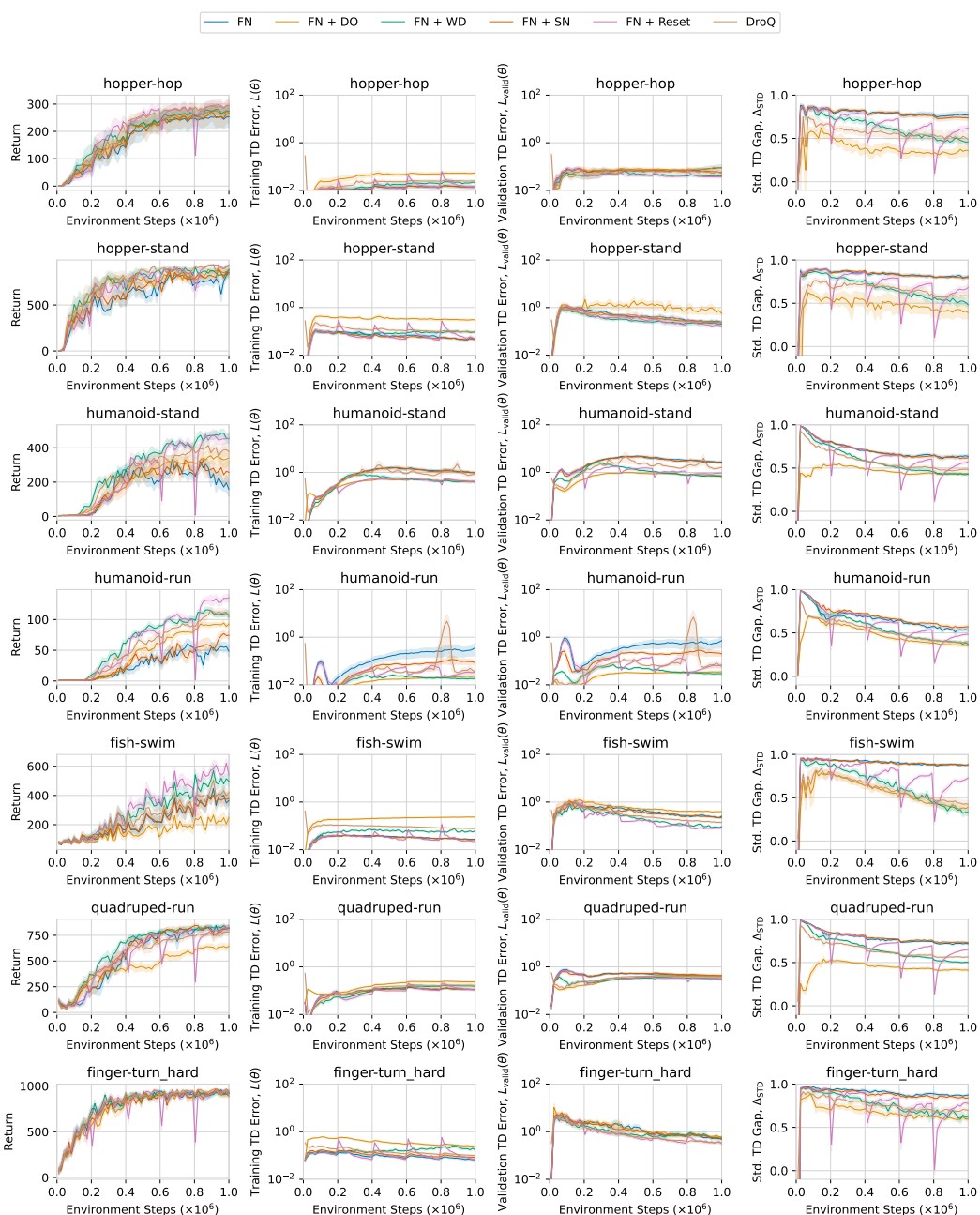

Figure 15: **The effects of different regularizations on 7 DMC tasks in the offline streaming setting.** All agents use feature normalization in the last layer to stabilize TD learning. On `fish-swim`, `humanoid-run` and `humanoid-stand`, the evaluation returns of different regularization approaches generally correlates well with their TD errors. On `hopper-hop`, `finger-turn_hard` and `hopper-stand`, no obvious correlation can be seen as all of approaches perform quite similarly. Specifically, on `fish-swim`, the top performing method correlates better with the validation TD error compared to the training TD error.

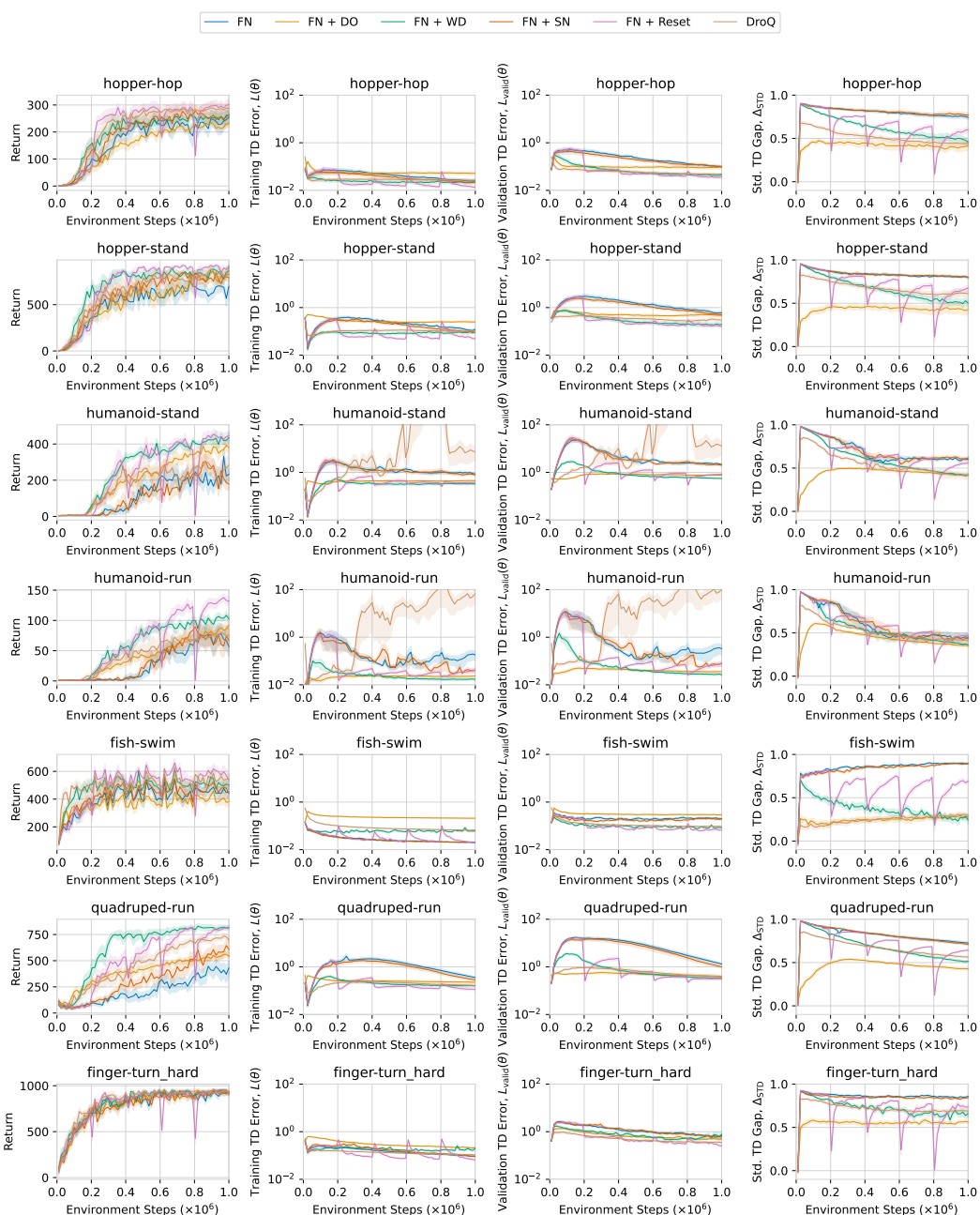

Figure 16: **The effects of different regularizations on 7 DMC tasks in the offline shuffled streaming setting.** All agents use feature normalization in the last layer to stabilize TD learning. On `humanoid-stand`, `humanoid-run`, `quadruped-run` and `fish-swim`, the top performing methods tend to have lower TD error. Specifically, on `fish-swim`, the top performing method correlates better with the validation TD error compared to the training TD error.

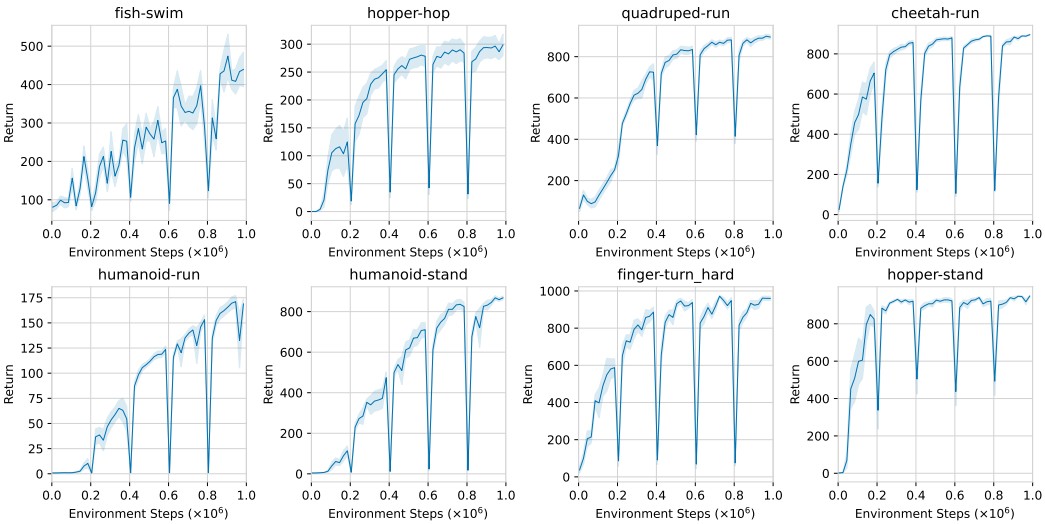

Figure 17: The data collecting policy for the offline analysis. The online training RL agent is a standard SAC with UTD ratio of 9 and gets periodically reset after every 200K steps.

## B.2 Full Results for AVTD

**Aggregated performance computation** To compute the aggregated performance for each method, we use the following protocol. For each environment, we normalize the return by the best average return achieved on each task (taking the maximum over all agent and all environment steps and the average over all eight seeds). After obtaining the normalized return for each environment, method and seed, we aggregate them over nine DMC tasks and four Gym tasks to obtain the aggregated normalized score.

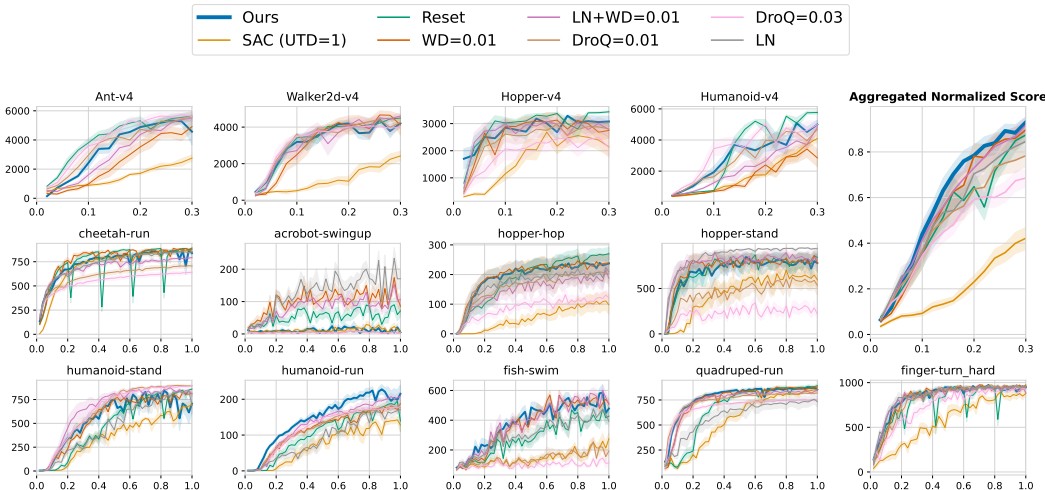

Figure 18: AVTD compared to different regularizers and the standard SAC baseline with UTD=1. For Gym tasks, UTD=20 is used and the actor is updated once per 20 critic updates. For DMC tasks, UTD=9 is used and the actor is updated with every critic update.

