# OpenReview forum: "Efficient Deep Reinforcement Learning Requires Regulating Statistical Overfitting"
_NeurIPS.cc/2022/Workshop/Offline_RL — Offline RL Workshop NeurIPS 2022_

### Official Review · Reviewer_HbqH · 2022-10-13
**A promising way to introduce the notion of cross validation in deep RL**

**Rating:** 3
**Confidence:** 4

**Review:**

The work investigates the failure of deep RL algorithms in the high update-to-data ratio regime. The authors find the overfitting of the TD error may be the culprit. Based on this observation, new algorithm that performs model selection online according to the validation TD error is proposed. The efficacy of the new algorithm is backed by the empirical results.

Though I in general like this work, it is worth mentioning that this submission significantly exceeds the page limit. As a result, I have to recommend rejection.

---

### Official Review · Reviewer_Z3Gk · 2022-10-18

**Rating:** 2
**Confidence:** 5

**Review:**

This paper violates workshop submission instructions to have only 4-5 pages of content. This paper has 9-pages content.
To be fair to other submissions, I'd recommend desk rejecting this paper.